# Progress in Probe-Based Sensing Techniques for In Vivo Diagnosis

**DOI:** 10.3390/bios12110943

**Published:** 2022-10-31

**Authors:** Cheng Zhou, Zecai Lin, Shaoping Huang, Bing Li, Anzhu Gao

**Affiliations:** 1Institute of Medical Robotics, Shanghai Jiao Tong University, Shanghai 200240, China; 2Department of Automation, Shanghai Jiao Tong University, Shanghai 200240, China; 3Department of Biomedical Engineering, Shanghai Jiao Tong University, Shanghai 200240, China; 4Institute for Materials Discovery, University College London, London WC1E 7JE, UK

**Keywords:** minimally invasive surgery, endoluminal intervention, physical sensing, biochemical sensing, in vivo diagnosis

## Abstract

Advancements in robotic surgery help to improve the endoluminal diagnosis and treatment with minimally invasive or non-invasive intervention in a precise and safe manner. Miniaturized probe-based sensors can be used to obtain information about endoluminal anatomy, and they can be integrated with medical robots to augment the convenience of robotic operations. The tremendous benefit of having this physiological information during the intervention has led to the development of a variety of in vivo sensing technologies over the past decades. In this paper, we review the probe-based sensing techniques for the in vivo physical and biochemical sensing in China in recent years, especially on in vivo force sensing, temperature sensing, optical coherence tomography/photoacoustic/ultrasound imaging, chemical sensing, and biomarker sensing.

## 1. Introduction

Surgical procedures can be divided into three main categories: open surgery, minimally invasive, and non-invasive surgery. Minimally invasive surgery (MIS), implementing the procedure through small incisions or natural orifices, has developed rapidly in recent decades, and continues to expand in many indications, such as cardiovascular, pancreatic, gastric, and endometrial cancers [1]. For example, the number of minimally invasive cardiovascular surgeries in 2019 in China reached 51,354, a 24.0% increase from 41,430 the year before [2]. Especially with the emergence of robotic surgery, minimally invasive surgery techniques relieve the patient’s pain greatly, because of the shorter operation time, lower risk, small incision, and more precise medicine [3,4,5].

The structures of commercial needle [6], guidewire [7], catheter [8], balloon [9], continuum robots [10,11,12,13], and other medical instruments, whose outer diameter is down to several millimeters, submillimeters, or even micro-millimeters, are naturally fit for minimally invasive surgery due to their slender characteristics (typical lumens or cavities of humans are shown in Figure 1). Especially after introducing continuum robots to surgery, diagnosis or treatment can be introduced deep into the human anatomy after passing through tortuous lumens [14,15].

During the operation procedure, surgeons cannot recognize the precise status of the robots in an uncertain environment without the sensing capability. To improve surgical performance, medical robots or devices can be mounted with sensors to collect interaction or surrounding information, including contact force [16,17,18], surrounding temperature, vision [19,20], and geometrics or pathology inside the vessels [21,22,23]. Additionally, to detect diseases in their early stages before symptoms appear, it is significant to make deep tissue diagnoses. Biochemical sensing plays an important role in early diagnosis. When tissues produce adverse effects and pathological changes, the body release important chemical signals or biomarkers, such as pH [24,25,26], superoxide anion [27], glucose [28,29,30,31], specific proteins [32], and H_2_O_2_ [33]. The expression of these signals often precedes the noticeable decline in tissue or organ function.

In this paper, we focus on probe-based biosensors (as shown in Figure 2), including sensors with “probe” shapes, such as needle-based, guidewire-based, catheter-based, balloon-based, and continuum robot-based sensors. We review these sensors from the perspectives of in vivo physical sensing techniques, including force sensing, temperature sensing, optical coherence tomography (OCT) imaging, photoacoustic (PA) imaging, and ultrasound (US) imaging, and biochemical sensing techniques developed by domestic researchers in recent years. Based on this review, we aim to provide a different perspective focused on interventional surgery. This will support the optimization of the sensors’ structural design and surface modification, and improve the devices’ functionality and integration level.

## 2. Probes for In Vivo Physical Sensing

### 2.1. Probes for In Vivo Force Sensing

Visual feedback is a basic function for surgeons in minimally invasive operations, but this single-mode feedback is not sufficient for delicate manipulation. Force sensing can provide intuitive, real-time, and interactive feedback to the operator. Therefore, it plays an important role in the high-quality diagnosis of biomedical environment [34,35].

#### 2.1.1. Fiber Bragg Grating-Based Force Sensor

Fiber Bragg grating (FBG) is a periodic grating with a different refractive index to the core of silica optical fiber. The refractive index of periodic FBG is usually modified by UV exposition. The spatial period of the grating, as well as the refractive indices of the core, are quantities that depend on mechanical strain and temperature, resulting in the change of detected reflected wavelength. This can be used for accurate axial micro-strain measurements at high sampling rates in real time [36,37]. The principle of FBG for force sensing is shown in Figure 3.

Optical fiber is a common optical waveguide for the transmission of light signals [38]. FBG-based optical fiber sensors have many advantages, such as high flexibility, lightweight, dielectric suitability, and MRI compatibility. Additionally, owing to optical fiber’s inherent vantages that include miniaturized size, biocompatibility, and intrinsic sensing elements, it is suitable for fragile or confined environments, such as the intra-body parts of humans.

FBG sensors have been widely employed as surgical tools and biosensors, showing a great potential for biomedical engineering [39]. Flexible probes allow performing large-area tissue scanning or palpation for early stage cancer screening, which need to provide a gentle contact between the probe and tissue. During this procedure, a continuum manipulator equipped with a force/torque sensor is necessary [40]. Comparative studies between medical imaging-based visual operation and tactile force feedback-assisted operation reveal the superiority of interventional therapy with tactile force feedback [41]. The availability of tactile force feedback could reduce perioperative complications. Wang et al. proposed a scanning device for intraoperative thyroid gland endomicroscopy with one degree of freedom (DOF) FBG force sensor [42]. To obtain more information for clinical surgery, force-sensing devices with more degrees of freedom have been studied. Ping et al. presented a 3-DOF scanning device including an axial linear motion for approaching the tissue and two bending DOFs for surface scanning. To adapt the structure of commercial platforms, such as a standard endoscope, four FBG sensors were integrated into a 2.7 mm continuum robot. Three FBG sensors measured the transverse forces, while the other FBG sensor monitored the axial forces. Then, the flexible instrument with the FBG sensors was used for gastric endomicroscopy [43], as shown in Figure 4a. To avoid the failure of FBG, five single-mode fibers were buried into the deformable matrix in parallel rather than being simply pasted on the surface of robot. Ex vivo tissue palpation was implemented to validate the effectiveness of the sensor, and surface reaction forces and hard inclusions could be identified. In addition to producing force feedback for localizing tissue hard inclusions, an FBG-based fiber matrix was employed to reconstruct the surface profile of tissues during the process of palpation [44], as shown in Figure 4b. The moment can also influence the accuracy of sensing. The use of flexures coupled with FBG sensors has been demonstrated with high accuracy and repeatability for tissue force sensing. So, in Gao’s work, a decoupling sensitivity matrix based on beam theory was presented to analyze the tip force and moment [45], as shown in Figure 4c.

In addition to tip force for palpation, lateral contact between the continuum robot body and the surrounding environment is unavoidable. The calculation of the distributed strain along the fiber body may provide diagnoses of diseases, such as motility of the gastrointestinal tract. Zhang et al. proposed a distributed hyperelastic elastomer-packaged pressure sensor for lateral force sensing [46].

However, temperature noise is a common negative interference for fiber-optical sensing. To reduce the thermal noise effect on biosensors, Ran et al. employed a single microfiber Bragg grating-based biosensor with second and third harmonic resonance. During the heating process, the third harmonic resonance held a distinct response with respect to the second harmonic resonance, and thus the thermal noise can be decoupled [47].

#### 2.1.2. Electrical-Based Force Sensing

Electrical-based force sensing has the longest history of development and the most widespread application. A variety of electrical-based force sensors based on triboelectric nanogenerator, capacitive sensors, piezoresistive sensors, strain gauges, etc., have been attempted for minimally invasive surgical instruments. The capacitive theory is one of the most common methods used for force sensing. Senthil et al. proposed a stretchable capacitive-based pressure sensor patch, which can be integrated onto balloons towards continuous intra-abdominal pressure monitoring [48], for example, as shown in Figure 4d. Triboelectric nanogenerator, as a kind of new developing technology, and its application in force sensing has attracted more and more attention. Liu et al. reported an endocardial pressure sensor based on a triboelectric nanogenerator, which is not only flexible but also self-powered [8]. The sensor was assembled into a surgical catheter for minimally invasive surgery (MIS), and the endocardial pressure was monitored by using the changes in voltage. To sense the grasping force of surgical robots, some efforts need to be made at the integration of electrical-based sensors and small grippers. Hou et al. developed a biocompatible piezoresistive triaxial force sensor chip, which was integrated into the grip of a continuum robot to sense the grasping force of the MIS [49]. Yu et al. presented surgical forceps that consisted of double grippers with 3D pulling and grasp force sensing, and a simple structure with a double E-type strain beam was used as the substrate to minimize the size of the robot [50,51]. The main principle of these force sensing devices was to transform the voltage changes in the strain gauges into the force with a mathematical model. Machine learning is also used in electrical-based force sensing. Shi et al. proposed a method of force detection for a surgical robot, where the 3D force of the end-effector was decomposed by using an elastic element with an orthogonal beam structure. Moreover, a machine algorithm was used to learn the relationship between the acting force and the output voltage [52]. Other researchers also achieved the force sensing of robotics MIS with electromagnetic sensors, but most of the robots integrated with these sensors were hard to miniaturize, due to the rigid body or large diameter of the sensors.

#### 2.1.3. Other Techniques

Two-photon printing is a powerful technique to fabricate micro- or nanostructures for force sensing. With the continuous miniaturization of surgical instruments, force sensors based on micro or nanostructures have gradually become a research hotspot. Zou et al. proposed a Fabry–Perot-based nanonewton-scale force sensor, which was printed on a single-mode fiber tip to measure the adhesion forces applied on the surfaces of micro/nanoscale structures [53]. A combination of multiple manufacturing methods can provide a possible for complex microstructure to sensing. Li et al. exploited micro-3D fabrication technology combining two photon polymerization and carbon-nanotube spraying techniques to construct a microspring-based electrical resistive sensor on the tip of a continuum robot. To demonstrate its potential, the device was employed to monitor human arterial pulses and real-time non-invasive intraluminal intervention [54], as shown in Figure 4e.

### 2.2. Probes for In Vivo Temperature Sensing

Thermal therapy, such as laser ablation and optogenetics, is a commonly surgical procedure in precision medicine. Ablation, especially laser treatment, is a medical procedure to remove diseased tissue and optogenetics is a method to study the relationship between biological conduction and light. To prevent the tissue from overheating during the process of the aforementioned operation, tissue temperature needs to be accurately monitored to ensure the quality and efficiency of therapy.

In order to achieve this, a flexible device, which contains platinum microsensors with a linear temperature–resistance relationship and flexible interconnects, was attached to the surface of the medical cryoballoon to detect the temperature distribution. Platinum (Pt) was chosen due to its properties of biocompatibility and thermal-resistance linearity. The temperature sensing ability of the device was verified by an ex vivo porcine heart cryoablation experiment [55], as shown in Figure 5a. Additionally, a silicon-based probe with Pt-based thin-film thermal-resistance sensor was proposed to reach deep tissues [56], as shown in Figure 5b.

Franz et al. reported the use of blackbody radiation in the short-wave infrared range for the tissue temperature monitoring during the laser vaporization. The devices integrated with the catheter to allow temperature sensing in vivo [57]. Ding et al. presented the optoelectronic devices to achieve an efficient NIR-to-visible upconversion for thermal detection, featuring high sensitivities and low-power excitation. Furthermore, the thermal sensors can be assembled as arrays to map spatial temperature, and an integrated optical fiber-thermometer device was employed for monitoring temperature variations in the deep brain of mice [58]. An optical fiber probe composed of FBG and a graphene oxide film coated S-shape fiber taper was made into a reflection type. The temperature measurements were realized by monitoring the characteristic wavelength shifts of the SFT and FBG in the reflection spectrum [59], as shown in Figure 5c. The master distributions of temperature in cardiac tissue during and after ablation an important to understand and implement this process. Koh et al. proposed an ultrathin and flexible needle-type system that could be inserted into the myocardial tissue in a minimally invasive way. It can be used to monitor the temperature in the transmural direction during the process of ablation. The measurement results exhibited excellent performance [60], as shown in Figure 5d.

### 2.3. Probes for In Vivo Imaging

#### 2.3.1. Optical Coherence Tomography Imaging

OCT is a non-invasive three-dimensional imaging technique based on echo technology. It uses low-coherence light to scan tissues and capture the optical backscatter from deep tissues. It can provide cross-sectional images below the tissue surface rather than the outer surface images provided by microscopy. Due to its non-invasive, deep penetration, and high-resolution characteristics, the OCT technique has been widely used for medical imaging [61,62,63]. Compared with other imaging techniques such as magnetic resonance imaging (MRI) and optical microscopy, OCT possesses a higher resolution. The resolution can be classified into axial and transverse resolutions. The axial resolution can be expressed by $\Delta z\;=\frac{2\ln2}{\pi }\;\frac{\lambda^{2}}{\Delta \lambda }$, determined by the center wavelength λ and the spectral bandwidth ∆λ. The transverse resolution can be expressed by $\Delta x\;=\frac{4\lambda }{\pi }\;\frac{f}{d}$, determined by the center wavelength λ, the effective focal length f of the focusing optics, and the spot size d on the objective lens [64].

An endoscope integrated with OCT, such as intravascular optical coherence tomography (IVOCT), is a significant clinical application for in vivo imaging in situ. It enables obtaining information from a constrained space, such as blood vessels. These spaces are hard to image using traditional methods, such as optical microscopy. The IVOCT is commonly composed of a light source, a light detector, and a gradient-index lens with a microprism at the distal end. The light is emitted from the light source, and then the distal lens directs the light beam, resulting in a focused output beam perpendicular to the catheter axis [65,66].

To better understand physical information of the blood vascular, such as blood pressure, blood flow, and vascular stenosis, Wang et al. combined an OCT catheter with FBG to acquire shape parameters and OCT images in real time to reconstruct the vascular model [67]. Kang et al. demonstrated an all-fiber-based proximal-driven OCT catheter, in which the lens can directly be assembled with the commercial single-mode fiber [68]. The in vivo imaging capability of the catheter was evaluated in animal models. Li et al. presented a tri-modality intravascular imaging system that integrated a tri-modality probe, including OCT, ultrasound, and fluorescence imaging, which makes it promising for clinical management [69], as shown in Figure 6a.

#### 2.3.2. Ultrasound Imaging

Ultrasound imaging, which is a safe, effective, and inexpensive technique for continuous monitoring in vivo, has been widely used in biomedical applications, particularly in clinical diagnosis, including extracranial steno-occlusive lesions diagnosis, intracranial stenosis diagnosis, and acute intracranial occlusion [70,71,72,73]. Especially in the process of stent graft placement, the use of intra-arterial is necessary. With the advancements in endovascular treatment and device fusion, the use of intravascular ultrasound incorporated with other techniques can treat more complex vascular pathologies [74,75]. Catheter-based intravascular ultrasound is a commonly used method to obtain real-time sufficient geometrical and pathological information from inside vessels for auxiliary diagnosis of intravascular diseases [76]. There are two typical catheter-based transducers: a mechanically rotating single-element transducer and an electronically phased array transducer. Zhang et al. focused on developing all-optical ultrasound probe for a miniature imaging system. They presented an ultrasound generator based on a step-indexed multimode fiber coated with a carbon nanotube and silicone rubber. The ultrasound was excited at the composite membrane with a nanosecond pulsed laser. Additionally, the ultrasound detector was made with rare-earth-doped fiber incorporating two reflective Bragg reflectors. The ultrasound probe was evaluated by imaging the cross-sectional structure of a swine trachea ex vivo [77], as shown in Figure 6b. To measure multiple parameters in arteries, Hong et al. designed a dual-mode ultrasound imaging catheter, including forwarding-looking and side-looking transducer. The transducers were composed of a piezoelectric layer, a top matching layer, and a backing layer. Tissue phantom experiments indicated that the dual-mode catheter could be possibly used for a one-time acquisition of multiple parameters, such as morphological and functional flow information about the vessel [78]. For calculating fractional flow reserve, a method combining coronary (X-ray) angiography curvature images with the intravascular ultrasound cross-section images was used to reconstruct the 3D structure. Based on the constructed structure, hemodynamics analysis was used to calculate fractional flow reserve. Aiming to obtain compact driving system, Wang et al. utilized miniature rotary–linear ultrasonic motor to drive the imaging catheter without any transmission structures. The motor could realize rotational and linear movements by changing ultrasonic motor modes. This work holds great promise for further compact system design [79].

#### 2.3.3. Photoacoustic Imaging

Photoacoustic imaging (PAI), which makes high-resolution deep tissue visualization possible, is a fast-growing technology for medical diagnosis, such as cancer imaging [80,81]. The tissue absorbs laser energy instantaneously, when irradiated by pulse laser, and expands to produce ultrasonic waves. The ultrasonic signal is then received by sensors for post-processing. Photoacoustic imaging includes tomography and microscopic imaging. The former uses diffused light to stimulate tissues, resulting in an imaging depth of several centimeters and a spatial resolution of tens of microns. The latter uses a focused laser to irradiate biological tissue and detects ultrasonic signals, resulting in transverse resolution up to the submicron level [82].

Fiber integrated with a PAI device was used to image biological tissue at a subcellular resolution in a minimally invasive manner [83]. Photoacoustic imaging microscopy (PAM) excites a focused pulsed laser beam and detects ultrasound waves through a piezoelectric transducer, and then generates images in a line. Moving the scanner could obtain two- or three-dimensional imaging [84]. In order to image in a constrained environment, steering imaging transducers will offer more applications. An ultrasound beam based on photoacoustic effect was excited by laser irradiating the gold nanocomposites, which was dip-coated on the tip of an optical fiber, and then a micro concave prism was fixed on the gold nanocomposites’ surface to focus the ultrasound waves. The ultrasound echo was received by a Fabry–Perot fiber optic sensor to reconstruct the ultrasound field. The potential of this minimally invasive diagnostics was implemented on an ex vivo porcine tissue experiment [85]. To conveniently identify the area of interest, miniature endoscopy combining photoacoustic microscopy and white-light microscopy was proposed. The white-light microscopy can guide the PAM to the region of interest before imaging [86], as shown in Figure 6c.

**Figure 6 biosensors-12-00943-f006:**
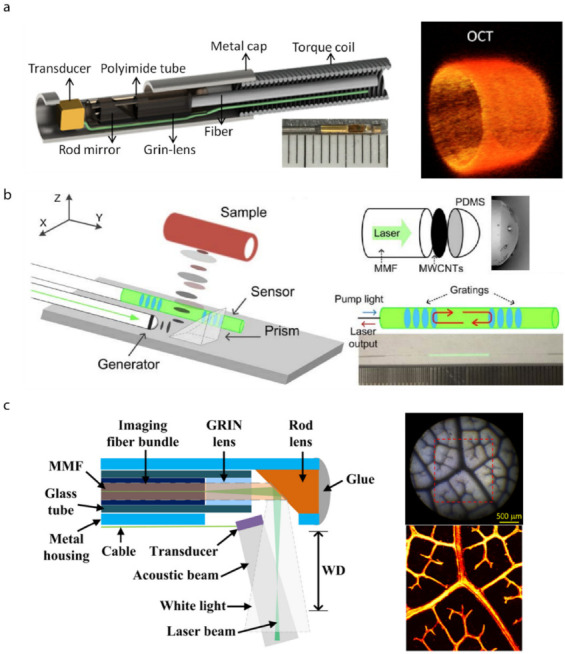
Probes for in vivo imaging. (**a**) Tri-modality probe and OCT image [69]. (**b**) The 125 um all-fiber-based ultrasound probe [77]. (**c**) Miniature probe-based photoacoustic microscopy and white-light microscopy endoscope. Reprinted with permission from Ref. [86]. Copyright 2020, John Wiley and Sons.

Catheter-based intravascular photoacoustic imaging is an emerging modality. It possesses many advantages. For example, the imaging depth of IVPA has been extended beyond the ballistic regime and IVPA can share the same detector with IVUS imaging, resulting in more complementary information of the tissue [87]. A miniature full field-of-view photoacoustic/ultrasonic endoscopic catheter system was used to depict the vasculature and morphology of the GI tract in vivo [88]. Cao et al. arranged a multimode fiber and acoustic device in a catheter tip to obtain efficient imaging overlap. The imaging capability was evaluated in a diseased porcine carotid artery and a human coronary artery ex vivo [87].

#### 2.3.4. Other Techniques

Raman spectroscopy is a vibrational spectroscopy capable of probing biomolecular information, which has wide applications in cell/tissue characterization and diagnosis without labeling. The development of the fiber-based Raman endoscopic probe makes imaging internal organs possible, such as diagnosing nasopharyngeal carcinoma [89]. An outer diameter 300-micron optical fiber with semispherical ball lens tip was inserted into a syringe needle for a deep-tissue Raman imaging [90].

The diagnostic value of probe-based confocal laser endomicroscopy (pCLE) has been recognized in many medical fields, such as endoscopic surveillance of Barrett’s esophagus [91], diagnosis of gastric carcinoma and precancerous lesions [92], and colon polyp histology [93]. Table 1 shows the comparison between these main in vivo imaging techniques [94].

## 3. Probes for In Vivo Biochemical Sensing

### 3.1. Probes for In Vivo Chemical Sensing

A chemical sensing system contains receptors for the target biomolecules in the tissue fluid, and the transducers to convert the results into measurable signals, such as electrical or optical signals [95,96]. Cancer is a common and intractable disease all over the world. There are many efforts devoted to defeating cancer. A rapid and effective diagnosis is significant. In the comparison of normal differentiated adult and cancer cells, the extracellular pH of the former is about 7.4, but cancer cells have a higher pH. Dysregulated pH is a well-known cancer indicator, which could be employed to diagnose cancer [97,98]. Optical fiber-based pH sensors have the ability to measure pH in deep tissues. Chen et al. proposed a miniaturized probe to detect the pH of the cancer tissue environment, based on the ratio fluorescence method. The 520 nm laser passed through the fiber and irradiated the fluorophore, which changed with the pH and the spectral band remained almost unchanged. The fluorescence emission light came back from the inner cladding, and then the returned light wave was detected by a spectrometer [24]. A U-shaped multimode optical fiber was demonstrated by Tang et al. The U-shaped bare region was coated with a hybrid organic–inorganic composite film, as a pH-sensitive layer. This bonding was affected by the hydrogen concentration in the solution, resulting in a refractive index change in the film. The sensor can be used to monitor the pH of human serum [25], as shown in Figure 7a. A hydrogel-based fiber pH sensor was demonstrated by Gong et al. The sensor was in situ photo-polymerized on the optical fiber tip with pH-sensitive hydrogel, the principle of which is the same as the fluorophore mentioned before. This optical fiber sensor was used to measure the pH of cancerous lung tissue [26], as shown in Figure 7b. To build fully biocompatible systems, Li et al. constructed in vivo red blood cells waveguide using two fiber probes in a microfluidic capillary to construct biosensors. By detecting the light propagation mode of the biosensor, the pH of blood can be detected in real time with high accuracy [99], as shown in Figure 7c. Peng et al. introduced a carbon-fiber microelectrode-based chemical sensor for monitoring the superoxide anion. The superoxide anion was acknowledged to be related to the development of many neurological diseases, including Alzheimer’s disease [27], as shown in Figure 7d.

### 3.2. Probes for In Vivo Biomarkers Sensing

Blood glucose concentration is a typical parameter to represent the human metabolic level. Chen et al. coated needle electrodes with polyaniline nanofiber, platinum nanoparticles, glucose oxidase enzyme, and porous layers with a layer-by-layer deposition process [28]. Nanoparticles incorporated into conductive polyaniline nanofibers resulted in a high surface-to-volume ratio for the immobilization of electrocatalytic glucose enzyme. The performance was then tested by inserting the needle into mice models, showing an excellent response to the concentration of blood glucose and good biocompatibility with the tissue. A minimally invasive glucose probe with an electropolymerized conductive polymer polyaniline core capable of continuously monitoring subcutaneous glucose has been developed [29]. In vivo experiments using mice models showed the real-time response to the variation of blood glucose, as shown in Figure 8a. Additionally, a chitosan/sodium-alginate-modified polysulfone hollow fibrous membrane was fixed on the stainless-steel needle electrode [30]. The needle electrode can be inserted into the skin to record responsive currents to detect blood glucose, as shown in Figure 8b. With the help of advanced manufacturing techniques, a Fabry–Perot cavity sensor was printed on the tip of a single-mode optical fiber by two-photon printing for glucose detection. The sensor was sensitive to the refractive index changes induced by the concentration changes in glucose [31], as shown in Figure 8c.

Wang et al. demonstrated a new type of functionalized multi-walled carbon nanotubes twisted fiber bundles to monitor multiple disease biomarkers, such as ions and prostate-specific antigens, hydrogen peroxide, and glucose [100]. Zhang et al. developed a flexible microelectrode based on carbon fiber wrapped by gold-nanoparticle-decorated nitrogen-doped carbon nanotube arrays, and researched its clinical applications in detecting the biomarker H_2_O_2_ expressed by living cancer cells in situ [33], as shown in Figure 8d.

Surface plasmon resonance (SPR) possesses a high compatibility with fiber-optic techniques. The sensors are sensitive to the refractive index of the certain materials. SPR sensors are usually constructed by coating a metal film on fiber surfaces, where reactions occur between the film and the environment. Additionally, the reaction changes the complex refractive index of the medium near the sensor surface, and therefore the SPR condition. Integrating SPR devices with an optical fiber can empower the biosensing systems with easily reading and in vivo monitoring capability [101]. Guo et al. presented a biosensor by coating a nanometer-scale silver film on tilted fiber Bragg grating to detect urinary protein variations. The sensors have the potential for a narrow endoluminal intervention in vivo [101]. A rare-earth-modified photothermal FBG-based fiber with fiber-optic fluorescent sensor was proposed to detect tumors in vivo. The fiber probe can turn the 450 nm excitation laser to an echo wavelength of 550 nm under a hypoxia tumor microenvironment and then kill the tumor through the photothermal effect [102].

## 4. Conclusions

This review focused on probe-based sensing techniques for the endoluminal intervention of minimally invasive or non-invasive procedures. It aimed to help researchers in the field of in vivo sensing techniques to diagnose/treat diseases, manipulate/assist interventional medical tools, and understand the latest domestic progress in recent years. The structure of the probe with a miniature size naturally fits minimally invasive procedures and enables a deeper tissue operation. A large amount of physiological information leads to a variety of in vivo sensing technologies, including physical sensing and biochemical sensing, in recent years. Physical sensing is the most direct perception for surgical intervention. For force sensing, in order to better control surgical instruments and improve surgical safety, the force feedback between instruments and tissues is important for both surgeons and patients. For temperature sensing, the real-time monitoring of the tissue temperature can improve the effectiveness of ablation. Research on the pathological characteristics of vascular diseases and the morphology of vessels has attracted vast attention from clinical medical researchers. Optical coherence tomography imaging, ultrasound imaging, photoacoustic imaging, and other imaging techniques can penetrate tissues, such as the vessel wall, to image its deep morphology to provide precise diagnosis. Different from these physical sensing techniques, biochemical sensing plays a more important role in early diagnosis rather than surgical operation. When tissues produce adverse effects and pathological changes in the initial stage of a disease, some important chemicals or biomarkers, such as pH, glucose, protein, and H_2_O_2_, often precedes a detectable decline in function. Thus, biochemical sensing can reflect the development of the disease, but physical sensing cannot distinguish the differences very well. Table 2 shows the comparison between these main probe-based sensing techniques.

There are many kinds of force sensors, including but not limited to FBG-based sensors, triboelectric nanogenerators, capacitive sensors, piezoresistive sensors, and strain gauges. Some of them are stable, reliable, and easy to miniaturize, but have large modulus and single measurement modes, such as FBG-based sensors. Some of them are soft, sensitivity-adjustable, and self-powered, but their applications may be compromised by their unstable performance, difficulty of miniaturizing, and the difficulty to producing plentifully. As for in vivo temperature sensing, it can be used to monitor thermal effects to avoid overheating, but it has not been widely used in the clinical scenario and can easily be replaced by other technologies, such as visual-based sensors. For in vivo imaging, IVUS is the most commonly used method in intravascular imaging, and OCT possesses a high resolution of intravascular imaging and has been applied to ophthalmology and other surgeries. The US has a more extensive imaging range and can penetrate tissues 7–15 mm deeper, while the OCT only has a 1–2 mm penetration power. Considering the high resolution of light-based OCT imaging techniques, the OCT can easily be disturbed by the environment, such as the scattering process in the background. Thus, the combination of US and OCT imaging techniques can achieve macroscopic and microscopic pathological characteristics and morphology of the tissue. PA imaging combines the high-contrast characteristics of optical imaging and the high-penetration-depth characteristics of ultrasound imaging, which can provide high-resolution and high-contrast tissue imaging. PA’s imaging depth and spatial resolution in tissues are related to acoustic frequency [94]. For in vivo biochemical sensing, Doctors may obtain and evaluate symptoms in situ via markers most related to it, rather than judging the symptoms through indirect features that may lead to misjudgments. However, the lack of standard micro/nano fabrication approaches, long-term stability, and repeatability are still challenging problems for these sensors, especially when they are used in confined anatomical scenarios.

These probe-based sensing techniques for in vivo diagnosis do help the doctor to understand the patients’ conditions more quickly and directly. However, even though these probe-based in vivo sensing techniques are implemented in minimally invasive way, these operation procedures are still not easy to conduct. These problems further hinder the commercialization of probe-based sensors. However, through successive efforts, in vivo sensing is becoming more precise, more instantaneous, and easier.

## Figures and Tables

**Figure 1 biosensors-12-00943-f001:**
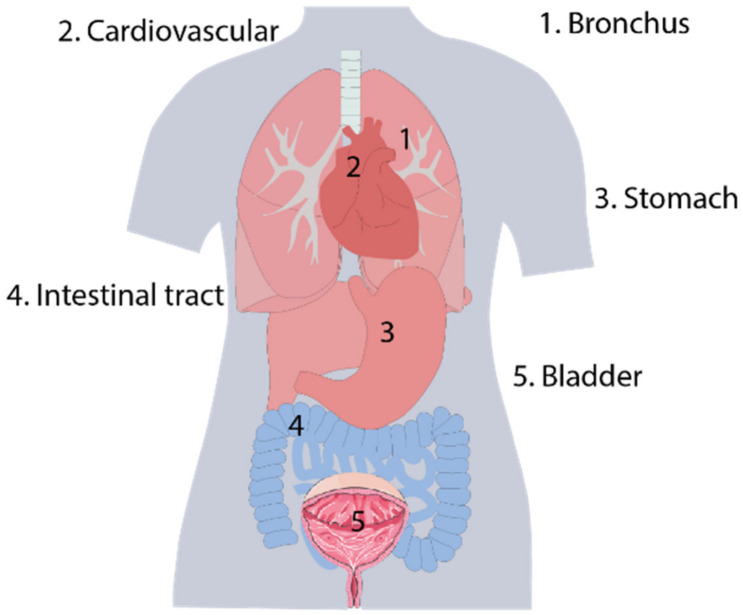
Typical lumens or cavities of humans.

**Figure 2 biosensors-12-00943-f002:**
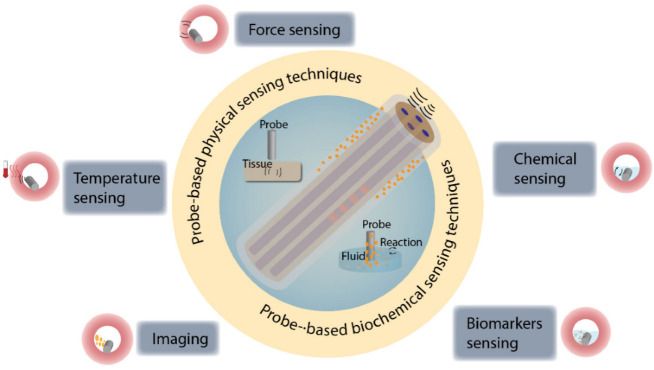
Overview of probe-based sensing techniques for in vivo diagnosis. Various probe-based sensors have been developed to detect physical and biochemical signals in vivo.

**Figure 3 biosensors-12-00943-f003:**
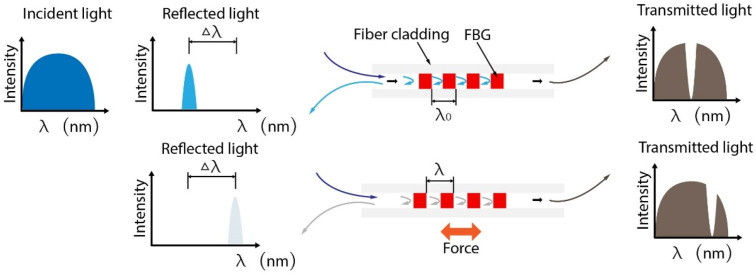
Mechanism of FBG-based force sensing. The FBG reflects the specific wavelength related to grating period, which can be changed with the axial strain. According to the physical characteristics and the reflected wavelength, the axis force can be calculated.

**Figure 4 biosensors-12-00943-f004:**
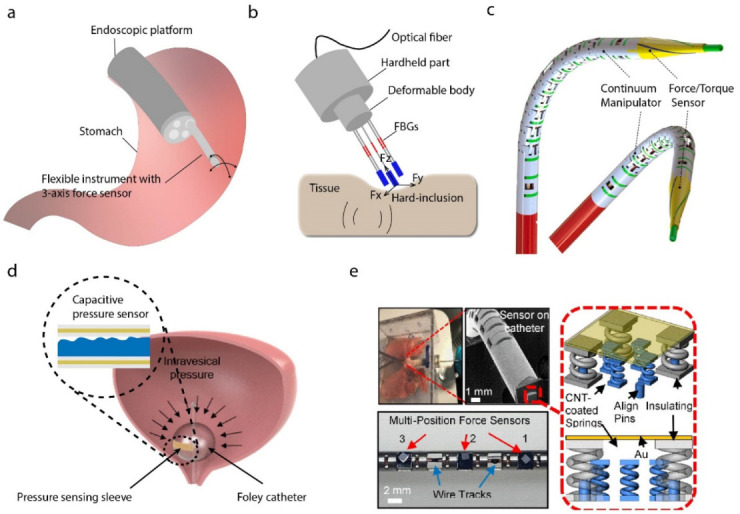
Probes for in vivo force sensing. (**a**) Flexible medical instrument for 3-axis force sensing [43]. (**b**) An elastic element with an orthogonal beam structure for end-effector 3D force decomposing [44]. (**c**) A spiral FBG force sensors-based method to measure the force and torque applied at the tip of the probe. Reprinted with permission from Ref. [45]. Copyright 2020, Elsevier. (**d**) A stretchable capacitive-based pressure sensor for continuous intra-abdominal pressure monitoring [48]. (**e**) Printed micro-spring with a carbon-nanotube force sensing layer on the tip of a continuum robot achieving a non-invasive intraluminal intervention. Reprinted with permission from Ref. [54]. Copyright 2019, American Chemical Society.

**Figure 5 biosensors-12-00943-f005:**
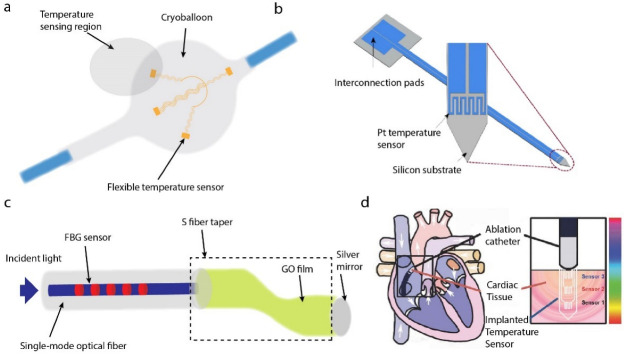
Probes for in vivo temperature sensing. (**a**) Flexible platinum-based temperature-sensing microsensors for cryoablation temperature monitoring [55]. (**b**) Pt-thermo-resistance-based temperature sensor for temperature monitoring to prevent overheating issues in optogenetics [56]. (**c**) Reflective FBG-based probe for temperature sensing [59]. (**d**) Ultrathin temperature sensor for cardiac ablation monitoring. Reprinted with permission from Ref. [60]. Copyright 2015, John Wiley and Sons.

**Figure 7 biosensors-12-00943-f007:**
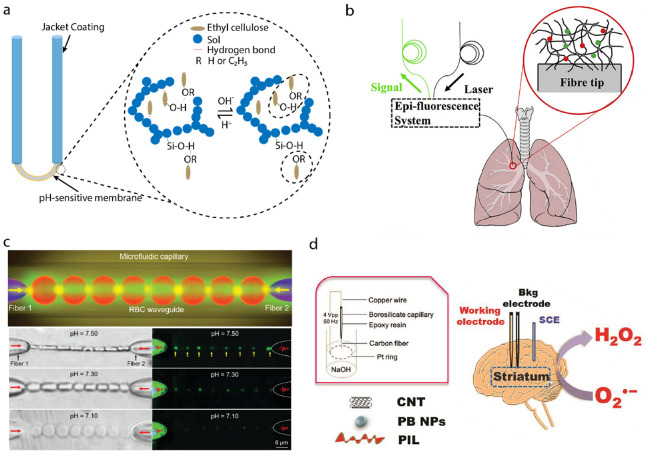
Probes for in vivo chemical sensing. (**a**) A U-shaped optical pH sensor based on hydrogen bonding [25]. (**b**) A hydrogel-based optical-fiber fluorescent pH sensor. Reprinted with permission from Ref. [26]. Copyright 2020, Elsevier. (**c**) Red-blood-cells waveguide for blood pH detection in real time. Reprinted with permission from Ref. [99]. Copyright 2019, John Wiley and Sons. (**d**) In vivo monitoring of superoxide anion with functionalized ionic liquid polymer-decorated microsensor. Reprinted with permission from Ref. [27]. Copyright 2019, Elsevier.

**Figure 8 biosensors-12-00943-f008:**
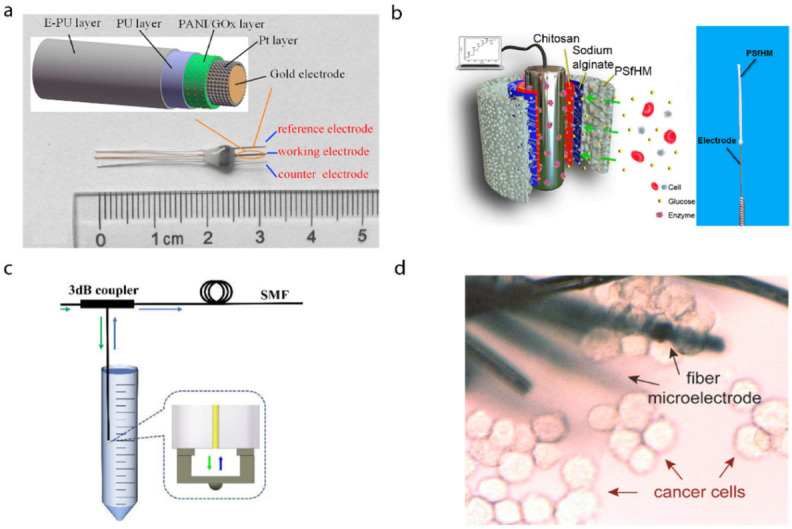
Probes for in vivo biomarkers sensing. (**a**) A needle-type blood glucose biosensor for long-term in vivo monitoring. Reprinted with permission from Ref. [29]. Copyright 2017, Elsevier. (**b**) Implantable glucose biosensing probe. Reprinted with permission from Ref. [30]. Copyright 2019, American Chemical. Society. (**c**) Fiber-tip micro Fabry–Perot interferometer for glucose concentration measurement. The Fabry–Perot interferometer is sensitive to the refractive index changes in analytes [31]. (**d**) Flexible nanohybrid microelectrode arrays for in situ biomarker H_2_O_2_ detection in live cancer cells. Reprinted with permission from Ref. [33]. Copyright 2018, Elsevier.

**Table 1 biosensors-12-00943-t001:** Comparison between in vivo imaging techniques.

Imaging Techniques	Axial Resolution	Transverse Resolution	Penetrating Depth	Integrated Size	Typical Applications
OCT imaging	0.2–1 μm [63]	0.6–2 μm [63]	1–2 mm [76]	Submillimeter [65,67,68]	Vascular shape reconstruction [67] Intracoronary optical coherence tomography [68]
US imaging	20–200 μm [72]	120–250 μm [76]	7–15 mm [76]	Submillimeter [77] Millimeter [78,79]	Intravascular imaging [74,78,79] Trachea imaging [77]
PA imaging	From sub-micrometer to sub-millimeter [94]	From sub-micrometer to sub-millimeter [94]	From sub-millimeter to depths up to several millimeters [94]	Submillimeter [85] Millimeter [73,75,86,87,88]	Tissue imaging [85,86,88] Intravascular imaging [87]

**Table 2 biosensors-12-00943-t002:** Comparison between probe-based sensing techniques.

Sensing Techniques	Sensing Mechanism	Carrier	Overall Diameter	Medical Scenario
Force sensing	Fiber Bragg grating sensors [42,43,44,45,46] Plastic Fiber Bragg grating sensors [36] Triboelectric nanogenerator [8] Piezoresistive [49] Piezoelectric Strain gauges [50,51,52] Capacitive pressure sensor [48] Closed-loop force control[37] Fiber-tip microforce sensor [53] Carbon nanotube-coated microsprings [54]	Continuum robot [43,45,49,54] Scanning device [42] Probe [44] Polymer package [46] Plastic optical fiber [36] Catheter [8] Gripper [50,51,52] Foley catheter balloon [48] Optical fiber [53]	Submillimeter [36,48,53] Millimeter [42,43,44,45,46,49,50,51,54] Centimeter [8,37,52]	Thyroidectomy [42] Gastric Endomicroscopy [43] Hard-inclusion Localization [44] Optical biopsy [45] In vivo pressure sensor [46] Blood pressure [36] Confocal laser endomicroscopy [37] Endocardial pressure monitoring [8] Three-dimensional force sensing for forceps [49,50,51,52] Intra-abdominal pressure monitoring [48] Measurement of interfacial adhesion force [53] Transcutaneous monitoring of human arterial pulses [54]
Temperature sensing	Thermo-resistance effect[55,56,60] Infrared-to-visible upconversion [58] Thermal expansion and thermal-optic effects [59] Short-wave infrared [57]	Balloon [55] Silicon-based probe [56] Needle-type polymer [60] Silica fiber [58] Optical fiber [59] Silica fiber [57]	Submillimeter [56,57,58,59] Millimeter [55,60]	Cryoablation [55] Optogenetic [56] Arrhythmias [60] Deep-brain thermal detection [58] Laser vaporization [57]
Optical coherence tomography imaging	Light scattering [62,67,68,69]	Catheter [62,67,68,69]	Submillimeter [67,68] Millimeter [69]	Intravascular Imaging [62,67,68,69]
Ultrasound imaging	Pulse-echo [74,77,78,79]	Catheter [74,78,79] Optical fiber [77]	Submillimeter [79] Millimeter [78]	Intravascular Imaging [74,78,79] Trachea imaging [77]
Photoacoustic imaging	Pulse-echo [85,86,87,88]	Catheter [86,88] Optical fiber [85]	Millimeter [86,87,88]	Tissue imaging [85,86,88] Intravascular imaging [87]
Chemical sensing	pH sensitivity of fluorophore [24,26] Polymer aggregation leads to refractive index changes [25] Enzymatic catalysis and electrochemical reactions [27]	Optical fiber [24,26] Organic–inorganic composite film-coated optical fiber [25] Carbon fiber microelectrode [27]	Submillimeter [24,25,26]	Discrimination of tumorous and normal tissues [24,26] Chronic wounds and/or fetal acidosis [25] Superoxide anion detection [27]
Biomarker sensing	Enzymatic catalysis and electrochemical reactions [28,29,30,33] Fabry–Perot (FP) cavity biosensor [31] Surface plasmon resonance [32]	Implantable electrode [28,29,30,33] Optical fiber [31,32]	0.005–0.03 mm [33] Submillimeter [28,31,32] Millimeter [29,30]	Blood glucose detection [28,29,30,31] Urinary protein detection [32] Cancer biomarker H_2_O_2_ detection [33]

## Data Availability

Not applicable.

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
