# Peer review of "Progress in Probe-Based Sensing Techniques for In Vivo Diagnosis"

_biosensors, 2022, doi:10.3390/bios12110943_

Round 1
Reviewer 2 Report
The authors gave a relatively comprehensive and detailed review of probe-based sensing techniques for in-vivo diagnosis with a thoughtful perspective. I suggest accepting it after minor revisions.
1. Please explain what probe-based means in the introduction. Besides probe-based sensing technology, are there any other sensing technologies applied to in-vivo diagnosis?
2. In 2.1.1 Fiber Bragg grating-based force sensor part, in the first paragraph, the introduction of optical fiber is a bit abrupt. It may be smoother to introduce the principle of FBG first and then the optical fiber, that is, put the second paragraph ahead.
3. In 2.1.2 Electrical-based force sensing part and 2.1.3 Other techniques part, the introduction here lacks logic and is just a simple listing of some examples. It is better to add some related words between the contexts to clarify the line of writing ideas.
4. In 2.3 Probes for in-vivo imaging, several in-vivo imaging techniques are introduced here including OCT imaging, US imaging, PA imaging, etc. Can you briefly explain the advantages and disadvantages of each method and how to choose the imaging method according to the actual application requirements?
5. A more detailed discussion of the challenges and future directions of probe-based sensing technology for in-vivo diagnosis would be better.
6. Please improve the resolution of Figure 8b.
7. Please check the style of the references and pay attention to the uniform style of the references.
